# The Impact of Artificial Intelligence on Data System Security: A Literature Review

**DOI:** 10.3390/s21217029

**Published:** 2021-10-23

**Authors:** Ricardo Raimundo, Albérico Rosário

**Affiliations:** 1ISEC Lisboa, Instituto Superior de Educação e Ciências, 1750-142 Lisbon, Portugal; ricardo.raimundo@iseclisboa.pt; 2Research Unit on Governance, Competitiveness and Public Policies (GOVCOPP), University of Aveiro, 3810-193 Aveiro, Portugal

**Keywords:** artificial intelligence, security, security of data, security systems

## Abstract

Diverse forms of artificial intelligence (AI) are at the forefront of triggering digital security innovations based on the threats that are arising in this post-COVID world. On the one hand, companies are experiencing difficulty in dealing with security challenges with regard to a variety of issues ranging from system openness, decision making, quality control, and web domain, to mention a few. On the other hand, in the last decade, research has focused on security capabilities based on tools such as platform complacency, intelligent trees, modeling methods, and outage management systems in an effort to understand the interplay between AI and those issues. the dependence on the emergence of AI in running industries and shaping the education, transports, and health sectors is now well known in the literature. AI is increasingly employed in managing data security across economic sectors. Thus, a literature review of AI and system security within the current digital society is opportune. This paper aims at identifying research trends in the field through a systematic bibliometric literature review (LRSB) of research on AI and system security. the review entails 77 articles published in the Scopus^®^ database, presenting up-to-date knowledge on the topic. the LRSB results were synthesized across current research subthemes. Findings are presented. the originality of the paper relies on its LRSB method, together with an extant review of articles that have not been categorized so far. Implications for future research are suggested.

## 1. Introduction

The assumption that the human brain may be deemed quite comparable to computers in some ways offers the spontaneous basis for artificial intelligence (AI), which is supported by psychology through the idea of humans and animals operating like machines that process information by devices of associative memory [1]. Nowadays, researchers are working on the possibilities of AI to cope with varying issues of systems security across diverse sectors. Hence, AI is commonly considered an interdisciplinary research area that attracts considerable attention both in economics and social domains as it offers a myriad of technological breakthroughs with regard to systems security [2]. There is a universal trend of investing in AI technology to face security challenges of our daily lives, such as statistical data, medicine, and transportation [3].

Some claim that specific data from key sectors have supported the development of AI, namely the availability of data from e-commerce [4], businesses [5], and government [6], which provided substantial input to ameliorate diverse machine-learning solutions and algorithms, in particular with respect to systems security [7]. Additionally, China and Russia have acknowledged the importance of AI for systems security and competitiveness in general [8,9]. Similarly, China has recognized the importance of AI in terms of housing security, aiming at becoming an authority in the field [10]. Those efforts are already being carried out in some leading countries in order to profit the most from its substantial benefits [9]. In spite of the huge development of AI in the last few years, the discussion around the topic of systems security is sparse [11]. Therefore, it is opportune to acquaint the last developments regarding the theme in order to map the advancements in the field and ensuing outcomes [12]. In view of this, we intend to find out the principal trends of issues discussed on the topic these days in order to answer the main research question: What is the impact of AI on data system security?

The article is organized as follows. In Section 2, we put forward diverse theoretical concepts related to AI in systems security. In Section 3, we present the methodological approach. In Section 4, we discuss the main fields of use of AI with regard to systems security, which came out from the literature. Finally, we conclude this paper by suggesting implications and future research avenues.

## 2. Literature Trends: AI and Systems Security

The concept of AI was introduced following the creation of the notion of digital computing machine in an attempt to ascertain whether a machine is able to “think” [1] or if the machine can carry out humans’ tasks [13]. AI is a vast domain of information and computer technologies (ICT), which aims at designing systems that can operate autonomously, analogous to the individuals’ decision-making process [14].In terms of AI, a machine may learn from experience through processing an immeasurable quantity of data while distinguishing patterns in it, as in the case of Siri [15] and image recognition [16], technologies based on machine learning that is a subtheme of AI, defined as intelligent systems with the capacity to think and learn [1].

Furthermore, AI entails a myriad of related technologies, such as neural networks [17] and machine learning [18], just to mention a few, and we can identify some research areas of AI:(I)Machine learning is a myriad of technologies that allow computers to carry out algorithms based on gathered data and distinct orders, providing the machine the capabilities to learn without instructions from humans, adjusting its own algorithm to the situation, while learning and recoding itself, such as Google and Siri when performing distinct tasks ordered by voice [19]. As well, video surveillance that tracks unusual behavior [20];(II)Deep learning constitutes the ensuing progress of machine learning, in which the machine carry out tasks directly from pictures, text, and sound, through a wide set of data architecture that entails numerous layers in order to learn and characterize data with several levels of abstraction imitating thus how the natural brain processes information [21]. This is illustrated, for example, in forming a certificate database structure of university performance key indicators, in order to fix issues such as identity authentication [21];(III)Neural networks are composed of a pattern recognition system that machine/deep learning operates to perform learning from observational data, figuring out its own solutions such as an auto-steering gear system with a fuzzy regulator, which enables to select optimal neural network models of the vessel paths, to obtain in this way control activity [22];(IV)Natural language processing machines analyze language and speech as it is spoken, resorting to machine learning and natural language processing, such as developing a swarm intelligence and active system, while mounting friendly human-computer interface software for users, to be implemented in educational and e-learning organizations [23];(V)Expert systems are composed of software arrangements that assist in achieving answers to distinct inquiries provided either by a customer or by another software set, in which expert knowledge is set aside in a particular area of the application that includes a reasoning component to access answers, in view of the environmental information and subsequent decision making [24].

Those subthemes of AI are applied to many sectors, such as health institutions, education, and management, through varying applications related to systems security. These abovementioned processes have been widely deployed to solve important security issues such as the following application trends (Figure 1):(a)Cyber security, in terms of computer crime, behavior research, access control, and surveillance, as for example the case of computer vision, in which an algorithmic analyses images, CAPTCHA (Completely Automated Public Turing test to tell Computers and Humans Apart) techniques [6,7,12,19,25,26,27,28,29,30,31,32,33,34,35,36,37,38];(b)Information management, namely in supporting decision making, business strategy, and expert systems, for example, by improving the quality of the relevant strategic decisions by analyzing big data, as well as in the management of the quality of complex objects [2,4,5,11,14,24,39,40,41,42,43,44,45,46,47,48,49,50,51,52,53,54,55,56,57,58,59,60];(c)Societies and institutions, regarding computer networks, privacy, and digitalization, legal and clinical assistance, for example, in terms of legal support of cyber security, digital modernization, systems to support police investigations and the efficiency of technological processes in transport [8,9,10,15,17,18,20,21,23,28,61,62,63,64,65,66,67,68,69,70,71,72,73];(d)Neural networks, for example, in terms of designing a model of human personality for use in robotic systems [1,13,16,22,74,75].

Through these streams of research, we will explain how the huge potential of AI can be deployed to over-enhance systems security that is in use both in states and organizations, to mitigate risks and increase returns while identifying, averting cyber attacks, and determine the best course of action [19]. AI could even be unveiled as more effective than humans in averting potential threats by various security solutions such as redundant systems of video surveillance, VOIP voice network technology security strategies [36,76,77], and dependence upon diverse platforms for protection (platform complacency) [30].

The design of the abovementioned conceptual and technological framework was not made randomly, as we did a preliminary search on Scopus with the keywords “Artificial Intelligence” and “Security”.

## 3. Materials and Methods

We carried out a systematic bibliometric literature review (LRSB) of the “Impact of AI on Data System Security”. the LRSB is a study concept that is based on a detailed, thorough study of the recognition and synthesis of information, being an alternative to traditional literature reviews, improving: (i) the validity of the review, providing a set of steps that can be followed if the study is replicated; (ii) accuracy, providing and demonstrating arguments strictly related to research questions; and (iii) the generalization of the results, allowing the synthesis and analysis of accumulated knowledge [78,79,80]. Thus, the LRSB is a “guiding instrument” that allows you to guide the review according to the objectives.

The study is performed following Raimundo and Rosário suggestions as follows: (i) definition of the research question; (ii) location of the studies; (iii) selection and evaluation of studies; (iv) analysis and synthesis; (v) presentation of results; finally (vi) discussion and conclusion of results. This methodology ensures a comprehensive, auditable, replicable review that answers the research questions.

The review was carried out in June 2021, with a bibliographic search in the Scopus database of scientific articles published until June 2021. the search was carried out in three phases: (i) using the keyword Artificial Intelligence “382,586 documents were obtained; (ii) adding the keyword “Security”, we obtained a set of 15,916 documents; we limited ourselves to Business, Management, and Accounting 401 documents were obtained and finally (iii) exact keyword: Data security, Systems security a total of 77 documents were obtained (Table 1).

The search strategy resulted in 77 academic documents. This set of eligible break-downs was assessed for academic and scientific relevance and quality. Academic Documents, Conference Paper (43); Article (29); Review (3); Letter (1); and retracted (1).

Peer-reviewed academic documents on the impact of artificial intelligence on data system security were selected until 2020. In the period under review, 2021 was the year with the highest number of peer-reviewed academic documents on the subject, with 18 publications, with 7 publications already confirmed for 2021. Figure 2 reviews peer-reviewed publications published until 2021.

The publications were sorted out as follows: 2011 2nd International Conference on Artificial Intelligence Management Science and Electronic Commerce Aimsec 2011 Proceedings (14); Proceedings of the 2020 IEEE International Conference Quality Management Transport and Information Security Information Technologies IT and Qm and Is 2020 (6); Proceedings of the 2019 IEEE International Conference Quality Management Transport and Information Security Information Technologies IT and Qm and Is 2019 (5); Computer Law and Security Review (4); Journal of Network and Systems Management (4); Decision Support Systems (3); Proceedings 2021 21st Acis International Semi Virtual Winter Conference on Software Engineering Artificial Intelligence Networking and Parallel Distributed Computing Snpd Winter 2021 (3); IEEE Transactions on Engineering Management (2); Ictc 2019 10th International Conference on ICT Convergence ICT Convergence Leading the Autonomous Future (2); Information and Computer Security (2); Knowledge Based Systems (2); with 1 publication (2013 3rd International Conference on Innovative Computing Technology Intech 2013; 2020 IEEE Technology and Engineering Management Conference Temscon 2020; 2020 International Conference on Technology and Entrepreneurship Virtual Icte V 2020; 2nd International Conference on Current Trends In Engineering and Technology Icctet 2014; ACM Transactions on Management Information Systems; AFE Facilities Engineering Journal; Electronic Design; Facct 2021 Proceedings of the 2021 ACM Conference on Fairness Accountability and Transparency; HAC; ICE B 2010 Proceedings of the International Conference on E Business; IEEE Engineering Management Review; Icaps 2008 Proceedings of the 18th International Conference on Automated Planning and Scheduling; Icaps 2009 Proceedings of the 19th International Conference on Automated Planning and Scheduling; Industrial Management and Data Systems; Information and Management; Information Management and Computer Security; Information Management Computer Security; Information Systems Research; International Journal of Networking and Virtual Organisations; International Journal of Production Economics; International Journal of Production Research; Journal of the Operational Research Society; Proceedings 2020 2nd International Conference on Machine Learning Big Data and Business Intelligence Mlbdbi 2020; Proceedings Annual Meeting of the Decision Sciences Institute; Proceedings of the 2014 Conference on IT In Business Industry and Government An International Conference By Csi on Big Data Csibig 2014; Proceedings of the European Conference on Innovation and Entrepreneurship Ecie; TQM Journal; Technology In Society; Towards the Digital World and Industry X 0 Proceedings of the 29th International Conference of the International Association for Management of Technology Iamot 2020; Wit Transactions on Information and Communication Technologies).

We can say that in recent years there has been some interest in research on the impact of artificial intelligence on data system security.

In Table 2, we analyze for the Scimago Journal & Country Rank (SJR), the best quartile, and the H index by publication.

Information Systems Research is the most quoted publication with 3510 (SJR), Q1, and H index 159.

There is a total of 11 journals on Q1, 3 journals on Q2 and 2 journals on Q3, and 2 journal on Q4. Journals from best quartile Q1 represent 27% of the 41 journals titles; best quartile Q2 represents 7%, best quartile Q3 represents 5%, and finally, best Q4 represents 5% each of the titles of 41 journals. Finally, 23 of the publications representing 56%, the data are not available.

As evident from Table 2, the significant majority of articles on artificial intelligence on data system security rank on the Q1 best quartile index.

The subject areas covered by the 77 scientific documents were: Business, Management and Accounting (77); Computer Science (57); Decision Sciences (36); Engineering (21); Economics, Econometrics, and Finance (15); Social Sciences (13); Arts and Humanities (3); Psychology (3); Mathematics (2); and Energy (1).

The most quoted article was “CCANN: An intrusion detection system based on combining cluster centers and nearest neighbors” from Lin, Ke, and Tsai 290 quotes published in the Knowledge-Based Systems with 1590 (SJR), the best quartile (Q1) and with H index (121). the published article proposes a new resource representation approach, a cluster center, and the nearest neighbor approach.

In Figure 3, we can analyze the evolution of citations of documents published between 2010 and 2021, with a growing number of citations with an R2 of 0.45%.

The h index was used to verify the productivity and impact of the documents, based on the largest number of documents included that had at least the same number of citations. Of the documents considered for the h index, 11 have been cited at least 11 times.

In Appendix A, Table A1, citations of all scientific articles until 2021 are analyzed; 35 documents were not cited until 2021.

Appendix A, Table A2, examines the self-quotation of documents until 2021, in which self-quotation was identified for a total of 16 self-quotations.

In Figure 4, a bibliometric analysis was performed to analyze and identify indicators on the dynamics and evolution of scientific information using the main keywords. the analysis of the bibliometric research results using the scientific software VOSviewe aims to identify the main keywords of research in “Artificial Intelligence” and “Security”.

The linked keywords can be analyzed in Figure 4, making it possible to clarify the network of keywords that appear together/linked in each scientific article, allowing us to know the topics analyzed by the research and to identify future research trends.

## 4. Discussion

By examining the selected pieces of literature, we have identified four principal areas that have been underscored and deserve further investigation with regard to cyber security in general: business decision making, electronic commerce business, AI social applications, and neural networks (Figure 4). There is a myriad of areas in where AI cyber security can be applied throughout social, private, and public domains of our daily lives, from Internet banking to digital signatures.

First, it has been discussed the possible decreasing of unnecessary leakage of accounting information [27], mainly through security drawbacks of VOIP technology in IP network systems and subsequent safety measures [77], which comprises a secure dynamic password used in Internet banking [29].

Second, it has been researched some computer user cyber security behaviors, which includes both a naïve lack of concern about the likelihood of facing security threats and dependence upon specific platforms for protection, as well as the dependence on guidance from trusted social others [30], which has been partly resolved through a mobile agent (MA) management systems in distributed networks, while operating a model of an open management framework that provides a broad range of processes to enforce security policies [31].

Third, AI cyber systems security always aims at achieving stability of the programming and analysis procedures by clarifying the relationship of code fault-tolerance programming with code security in detail to strengthen it [33], offering an overview of existing cyber security tasks and roadmap [32].

Fourth, in this vein, numerous AI tools have been developed to achieve a multi-stage security task approach for a full security life cycle [38]. New digital signature technology has been built, amidst the elliptic curve cryptography, of increasing reliance [28]; new experimental CAPTCHA has been developed, through more interference characters and colorful background [8] to provide better protection against spambots, allowing people with little knowledge of sign languages to recognize gestures on video relatively fast [70]; novel detection approach beyond traditional firewall systems have been developed (e.g., cluster center and nearest neighbor—CANN) of higher efficiency for detection of attacks [71]; security solutions of AI for IoT (e.g., blockchain), due to its centralized architecture of security flaws [34]; and integrated algorithm of AI to identify malicious web domains for security protection of Internet users [19].

In sum, AI has progressed lately by advances in machine learning, with multilevel solutions to the security problems faced in security issues both in operating systems and networks, comprehending algorithms, methods, and tools lengthily used by security experts for the better of the systems [6]. In this way, we present a detailed overview of the impacts of AI on each of those fields.

### 4.1. Business Decision Making

AI has an increasing impact on systems security aimed at supporting decision making at the management level. More and more, it is discussed expert systems that, along with the evolution of computers, are able to integrate systems into corporate culture [24]. Such systems are expected to maximize benefits against costs in situations where a decision-making agent has to decide between a limited set of strategies of sparse information [14], while a strategic decision in a relatively short period of time is required demanded and of quality, for example by intelligent analysis of big data [39].

Secondly, it has been adopted distributed decision models coordinated toward an overall solution, reliant on a decision support platform [40], either more of a mathematical/modeling support of situational approach to complex objects [41], or more of a web-based multi-perspective decision support system (DSS) [42].

Thirdly, the problem of software for the support of management decisions was resolved by combining a systematic approach with heuristic methods and game-theoretic modeling [43] that, in the case of industrial security, reduces the subsequent number of incidents [44].

Fourthly, in terms of industrial management and ISO information security control, a semantic decision support system increases the automation level and support the decision-maker at identifying the most appropriate strategy against a modeled environment [45] while providing understandable technology that is based on the decisions and interacts with the machine [46].

Finally, with respect to teamwork, AI validates a theoretical model of behavioral decision theory to assist organizational leaders in deciding on strategic initiatives [11] while allowing understanding who may have information that is valuable for solving a collaborative scheduling problem [47].

### 4.2. Electronic Commerce Business

The third research stream focuses on e-commerce solutions to improve its systems security. This AI research stream focuses on business, principally on security measures to electronic commerce (e-commerce), in order to avoid cyber attacks, innovate, achieve information, and ultimately obtain clients [5].

First, it has been built intelligent models around the factors that induce Internet users to make an online purchase, to build effective strategies [48], whereas it is discussed the cyber security issues by diverse AI models for controlling unauthorized intrusion [49], in particular in some countries such as China, to solve drawbacks in firewall technology, data encryption [4] and qualification [2].

Second, to adapt to the increasingly demanding environment nowadays of a world pandemic, in terms of finding new revenue sources for business [3] and restructure business digital processes to promote new products and services with enough privacy and manpower qualified accordingly and able to deal with the AI [50].

Third, to develop AI able to intelligently protect business either by a distinct model of decision trees amidst the Internet of Things (IoT) [51] or by ameliorating network management through active networks technology, of multi-agent architecture able to imitate the reactive behavior and logical inference of a human expert [52].

Fourth, to reconceptualize the role of AI within the proximity’s spatial and non-spatial dimensions of a new digital industry framework, aiming to connect the physical and digital production spaces both in the traditional and new technology-based approaches (e.g., industry 4.0), promoting thus innovation partnerships and efficient technology and knowledge transfer [53]. In this vein, there is an attempt to move the management systems from a centralized to a distributed paradigm along the network and based on criteria such as for example the delegation degree [54] that inclusive allows the transition from industry 4.0 to industry 5.0i, through AI in the form of Internet of everything, multi-agent systems and emergent intelligence and enterprise architecture [58].

Fifth, in terms of manufacturing environments, following that networking paradigm, there is also an attempt to manage agent communities in distributed and varied manufacturing environments through an AI multi-agent virtual manufacturing system (e.g., MetaMorph) that optimizes real-time planning and security [55]. In addition, in manufacturing, smart factories have been built to mitigate security vulnerabilities of intelligent manufacturing processes automation by AI security measures and devices [56] as, for example, in the design of a mine security monitoring configuration software platform of a real-time framework (e.g., the device management class diagram) [26]. Smart buildings in manufacturing and nonmanufacturing environments have been adopted, aiming at reducing costs, the height of the building, and minimizing the space required for users [57].

Finally, aiming at augmenting the cyber security of e-commerce and business in general, other projects have been put in place, such as computer-assisted audit tools (CAATs), able to carry on continuous auditing, allowing auditors to augment their productivity amidst the real-time accounting and electronic data interchange [59] and a surge in the demand of high-tech/AI jobs [60].

### 4.3. AI Social Applications

As seen, AI systems security can be widely deployed across almost all society domains, be in regulation, Internet security, computer networks, digitalization, health, and other numerous fields (see Figure 4).

First, it has been an attempt to regulate cyber security, namely in terms of legal support of cyber security, with regard to the application of artificial intelligence technology [61], in an innovative and economical/political-friendly way [9] and in fields such as infrastructures, by ameliorating the efficiency of technological processes in transport, reducing, for example, the inter train stops [63] and education, by improving the cyber security of university E-Gov, for example in forming a certificate database structure of university performance key indicators [21] e-learning organizations by swarm intelligence [23] and acquainting the risk a digital campus will face according to ISO series standards and criteria of risk levels [25] while suggesting relevant solutions to key issues in its network information safety [12].

Second, some moral and legal issues have risen, in particular in relation to privacy, sex, and childhood. Is the case of the ethical/legal legitimacy of publishing open-source dual-purpose machine-learning algorithms [18], the needed legislated framework comprising regulatory agencies and representatives of all stakeholder groups gathered around AI [68], the gendering issue of VPAs as female (e.g., Siri) as replicate normative assumptions about the potential role of women as secondary to men [15], the need of inclusion of communities to uphold its own code [35] and the need to improve the legal position of people and children in particular that are exposed to AI-mediated risk profiling practices [7,69].

Third, the traditional industry also benefits from AI, given that it can improve, for example, the safety of coal mine, by analyzing the coal mine safety scheme storage structure, building data warehouse and analysis [64], ameliorating, as well, the security of smart cities and ensuing intelligent devices and networks, through AI frameworks (e.g., United Theory of Acceptance and Use of Technology—UTAUT) [65], housing [10] and building [66] security system in terms of energy balance (e.g., Direct Digital Control System), implying fuzzy logic as a non-precise program tool that allows the systems to function well [66], or even in terms of data integrity attacks to outage management system OMSs and ensuing AI means to detect and mitigate them [67].

Fourth, the citizens, in general, have reaped benefits from areas of AI such as police investigation, through expert systems that offer support in terms of profiling and tracking criminals based on machine-learning and neural network techniques [17], video surveillance systems of real-time accuracy [76], resorting to models to detect moving objects keeping up with environment changes [36], of dynamical sensor selection in processing the image streams of all cameras simultaneously [37], whereas ambient intelligence (AmI) spaces, in where devices, sensors, and wireless networks, combine data from diverse sources and monitor user preferences and their subsequent results on users’ privacy under a regulatory privacy framework [62].

Finally, AI has granted the society noteworthy progress in terms of clinical assistance in terms of an integrated electronic health record system into the existing risk management software to monitor sepsis at intensive care unit (ICU) through a peer-to-peer VPN connection and with a fast and intuitive user interface [72]. As well, it has offered an AI organizational solution of innovative housing model that combines remote surveillance, diagnostics, and the use of sensors and video to detect anomalies in the behavior and health of the elderly [20], together with a case-based decision support system for the automatic real-time surveillance and diagnosis of health care-associated infections, by diverse machine-learning techniques [73].

### 4.4. Neural Networks

Neural networks, or the process through which machines learn from observational data, coming up with their own solutions, have been lately discussed over some stream of issues.

First, it has been argued that it is opportune to develop a software library for creating artificial neural networks for machine learning to solve non-standard tasks [74], along a decentralized and integrated AI environment that can accommodate video data storage and event-driven video processing, gathered from varying sources, such as video surveillance systems [16], which images could be improved through AI [75].

Second, such neural networks architecture has progressed into a huge number of neurons in the network, in which the devices of associative memory were designed with the number of neurons comparable to the human brain within supercomputers [1]. Subsequently, such neural networks can be modeled on the base of switches architecture to interconnect neurons and to store the training results in the memory, on the base of the genetic algorithms to be exported to other robotic systems: a model of human personality for use in robotic systems in medicine and biology [13].

Finally, the neural network is quite representative of AI, in the attempt of, once trained in human learning and self-learning, could operate without human guidance, as in the case of a current positioning vessel seaway systems, involving a fuzzy logic regulator, a neural network classifier enabling to select optimal neural network models of the vessel paths, to obtain control activity [22].

### 4.5. Data Security and Access Control Mechanisms

Access control can be deemed as a classic security model that is pivotal do any security and privacy protection processes to support data access from different environments, as well as to protect unauthorized access according to a given security policy [81]. In this vein, data security and access control-related mechanisms have been widely debated these days, particularly with regard to their distinct contextual conditions in terms, for example, of spatial and temporal environs that differ according to diverse, decentralized networks. Those networks constitute a major challenge because they are dynamically located on “cloud” or “fog” environments, rather than fixed desktop structures, demanding thus innovative approaches in terms of access security, such as fog-based context-aware access control (FB-CAAC) [81]. Context-awareness is, therefore, an important characteristic of changing environs, where users access resources anywhere and anytime. As a result, it is paramount to highlight the interplay between the information, now based on fuzzy sets, and its situational context to implement context-sensitive access control policies, as well, through diverse criteria such as, for example, following subject and action-specific attributes. In this way, different contextual conditions, such as user profile information, social relationship information, and so on, need to be added to the traditional, spatial and temporal approaches to sustain these dynamic environments [81]. In the end, the corresponding policies should aim at defining the security and privacy requirements through a fog-based context-aware access control model that should be respected for distributed cloud and fog networks.

## 5. Conclusion and Future Research Directions

This piece of literature allowed illustrating the AI impacts on systems security, which influence our daily digital life, business decision making, e-commerce, diverse social and legal issues, and neural networks.

First, AI will potentially impact our digital and Internet lives in the future, as the major trend is the emergence of increasingly new malicious threats from the Internet environment; likewise, greater attention should be paid to cyber security. Accordingly, the progressively more complexity of business environment will demand, as well, more and more AI-based support systems to decision making that enables management to adapt in a faster and accurate way while requiring unique digital e-manpower.

Second, with regard to the e-commerce and manufacturing issues, principally amidst the world pandemic of COVID-19, it tends to augment exponentially, as already observed, which demands subsequent progress with respect to cyber security measures and strategies. the same, regarding the social applications of AI that, following the increase in distance services, will also tend to adopt this model, applied to improved e-health, e-learning, and e-elderly monitoring systems.

Third, subsequent divisive issues are being brought to the academic arena, which demands progress in terms of a legal framework, able to comprehend all the abovementioned issues in order to assist the political decisions and match the expectations of citizens.

Lastly, it is inevitable further progress in neural networks platforms, as it represents the cutting edge of AI in terms of human thinking imitation technology, the main goal of AI applications.

To summarize, we have presented useful insights with respect to the impact of AI in systems security, while we illustrated its influence both on the people’ service delivering, in particular in security domains of their daily matters, health/education, and in the business sector, through systems capable of supporting decision making. In addition, we over-enhance the state of the art in terms of AI innovations applied to varying fields.

### Future Research Issues 

Due to the aforementioned scenario, we also suggest further research avenues to reinforce existing theories and develop new ones, in particular the deployment of AI technologies in small medium enterprises (SMEs), of sparse resources and from traditional sectors that constitute the core of intermediate economies and less developed and peripheral regions. In addition, the building of CAAC solutions constitutes a promising field in order to control data resources in the cloud and throughout changing contextual conditions.

## Figures and Tables

**Figure 1 sensors-21-07029-f001:**
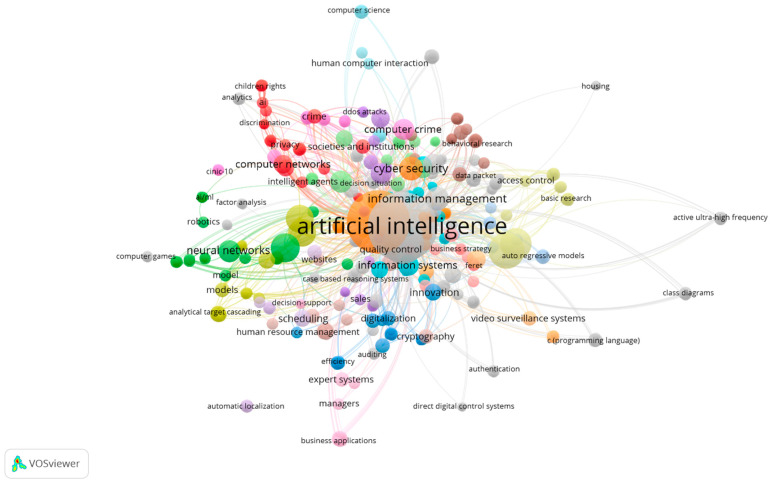
Subthemes/network of all keywords of AI—source: own elaboration.

**Figure 2 sensors-21-07029-f002:**
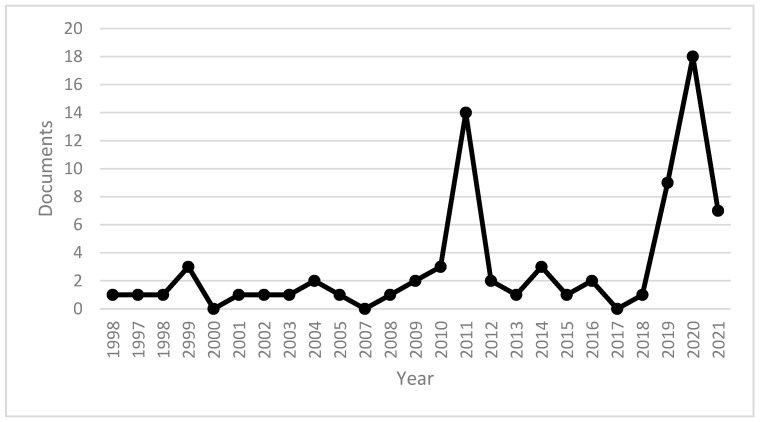
Number of documents by year. Source: own elaboration.

**Figure 3 sensors-21-07029-f003:**
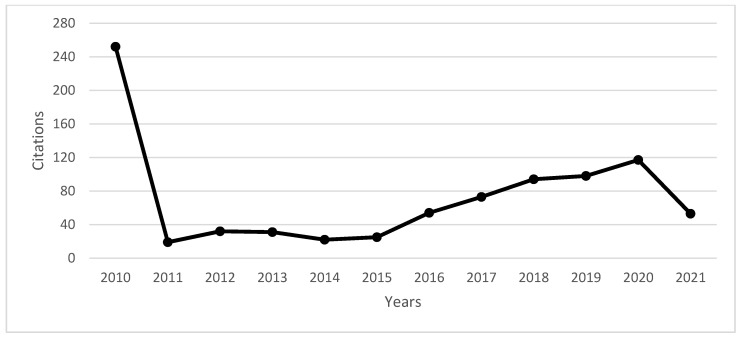
Evolution and number of citations between 2010 and 2021. Source: own elaboration.

**Figure 4 sensors-21-07029-f004:**
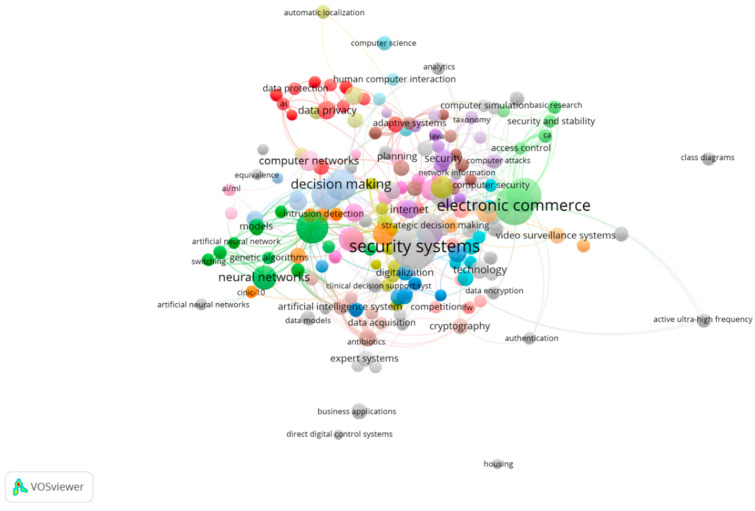
Network of linked keywords. Source: own elaboration.

**Table 1 sensors-21-07029-t001:** Screening methodology.

Database Scopus	Screening	Publications
Meta-search	Keyword: Artificial Intelligence	382,586
Inclusion Criteria	Keyword: Artificial Intelligence; Security	15,916
	Keyword: Artificial Intelligence; Security Business, Management, and Accounting	401
Screening	Keyword: Artificial Intelligence; Security Business, Management, and Accounting Exact Keyword: Security of Data; Security Systems Published until June 2021	77

Source: own elaboration.

**Table 2 sensors-21-07029-t002:** Scimago journal and country rank impact factor.

Title	SJR	Best Quartile	H Index
Information Systems Research	3.510	Q1	159
International Journal of Production Economics	2.410	Q1	185
Information and Management	2.150	Q1	162
Knowledge-Based Systems	1.590	Q1	121
Decision Support Systems	1.560	Q1	151
Industrial Management and Data Systems	0.990	Q1	103
Technology In Society	0.820	Q1	51
Computer Law and Security Review	0.820	Q1	38
Journal of the Operational Research Society	0.750	Q1	108
IEEE Transactions on Engineering Management	0.700	Q1	92
ACM Transactions on Management Information Systems	0.600	Q1	29
Journal of Network and Systems Management	0.490	Q2	35
Information and Computer Security	0.330	Q2	49
TQM Journal	0.540	Q2	67
IEEE Engineering Management Review	0.300	Q3	20
International Journal of Production Research	0.270	Q3	19
International Journal of Networking and Virtual Organizations	0.170	Q4	19
Electronic Design	0.100	Q4	7
Proceedings of the European Conference on Innovation and Entrepreneurship Ecie	0.130	-*	6
Icaps 2008 Proceedings of the 18th International Conference on Automated Planning and Scheduling	-*	-*	19
Wit Transactions on Information and Communication Technologies	-*	-*	13
Proceedings Annual Meeting of the Decision Sciences Institute	-*	-*	9
Proceedings of the 2014 Conference on IT In Business Industry and Government An International Conference By Csi on Big Data Csibig 2014	-*	-*	8
2nd International Conference on Current Trends In Engineering and Technology Icctet 2014	-*	-*	7
ICE B 2010 Proceedings of the International Conference on E Business	-*	-*	6
AFE Facilities Engineering Journal	-*	-*	2
2011 2nd International Conference on Artificial Intelligence Management Science and Electronic Commerce Aimsec 2011 Proceedings	-*	-*	-*
Proceedings of the 2020 IEEE International Conference Quality Management Transport and Information Security Information Technologies IT and Qm and Is 2020	-*	-*	-*
Proceedings of the 2019 IEEE International Conference Quality Management Transport and Information Security Information Technologies IT and Qm and Is 2019	-*	-*	-*
Proceedings 2021 21st Acis International Semi Virtual Winter Conference on Software Engineering Artificial Intelligence Networking and Parallel Distributed Computing Snpd Winter 2021	-*	-*	-*
Ictc 2019 10th International Conference on ICT Convergence ICT Convergence Leading the Autonomous Future	-*	-*	-*
2013 3rd International Conference on Innovative Computing Technology Intech 2013	-*	-*	-*
2020 IEEE Technology and Engineering Management Conference Temscon 2020	-*	-*	-*
2020 International Conference on Technology and Entrepreneurship Virtual Icte V 2020	-*	-*	-*
Facct 2021 Proceedings of the 2021 ACM Conference on Fairness Accountability and Transparency	-*	-*	-*
HAC	-*	-*	-*
Icaps 2009 Proceedings of the 19th International Conference on Automated Planning and Scheduling	-*	-*	-*
Information Management and Computer Security	-*	-*	-*
Information Management Computer Security	-*	-*	-*
Proceedings 2020 2nd International Conference on Machine Learning Big Data and Business Intelligence Mlbdbi 2020	-*	-*	-*
Toward the Digital World and Industry X 0 Proceedings of the 29th International Conference of the International Association for Management of Technology Iamot 2020	-*	-*	-*

Note: * data not available. Source: own elaboration.

## Data Availability

Not applicable.

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
