# Peer review of "The Impact of Artificial Intelligence on Data System Security: A Literature Review"

_sensors, 2021, doi:10.3390/s21217029_

Round 1

Reviewer 1 Report

Major changes:
1.    Paper uses just one database for the literaturÄ™ review. I suggest adding the literature review from other databases, not just Scopus.
2.    In line 17 is „secu-rity”, it should e „security”. In line 23 is „re-search” instead of „research”. This problem I also in the whole text. The paper must be read again and correct the word pagination.
3.    The paragraph (line 177-212) is not needed – it has no impact on the research. As I see the sorting method was by alphabet, so just one sentence informing about it is enough.
4.    In the Discussion chapter (line 285-287) authors mention that they identified five principal areas that have been underscored and deserve further investigation with regard to cyber systems security: cybersecurity in general, business decision making, electronic commerce business, AI social applications and neural networks. I suggest exclude „Cuber security in general”. If it is an area, the other areas are sub-areas, so there is just one main area or just four areas at a low level. It is not logical. What is more, I suggest to marking with different colour the main area in figure 4 – now the difference between these five areas and others is small. I think the treemaps chart will be better for this case – it will be more readable to see the main areas and it is possible to add the counter of papers or percentage of the total.
5.    If the suggestion from point 4 will be made, subchapter 5.1 is unnecessary. 
6.    In this kind of research, I suggest adding the graph with the presentation of the number of keywords/tags by each year. It will show how the cybersecurity problem grows in the last years.
7.    There is some problem with literaturÄ™, in the positions 21, 23, 33, 48, 54, 71 and 76 are „-„ after each author; in 46 is „…” before Sklar name,
Minor changes:
1.    In line 8 the „in further text” is not needed. It could be just artificial intelligence (IA).
2.    Figure 2 – It should be „Number of documents […]” not just „Documents […]”. Similar changes should be done in figure 3.

Author Response

Dear Reviewers,
We appreciate the criticisms and suggestions the changes were made directly in Word.´
Kind regards,

Reviewer 2 Report

This is an interesting topic to conduct a survey and the article covered a fair bit of existing state-of-the-art data security literature, including web domain, Ai techniques, decision making approaches, modelling methods and so on. The following suggestions could be taken into account to improve the writing and survey itself.

  • Please remove 'AI in further text', better to say AI.
  • Please remove 'review methodology' related content from abstract, e.g., 'The review entails 77 articles published in Scopus® database, presenting up-to-date 21 knowledge on the topic'. It is better to have a separate section titles 'Motivation and Methodology of this Survey' as Section 2. The authors could clearly motivate general audiences - why we need this new survey, why it is timely to do now and the methodology of this survey - what databases are considered, how, and the timeline.
  • This is awkward to cite like 'Figure 1: ... Source: own elaboration.'. Please provide formal citations and include URLs in the reference list. An interesting topic area have not been considered. It is necessary to consider 'Data Security and Access Control' direction and cover relevant papers. A few of the important papers can be included here. It would be great if the authors could discuss dynamic security policies (e.g., dynamic context-aware access control policies – cloud-based, fog-based, centralized or decentralized, based on the domains) and provide several future research directions, especially focusing – how the data security systems and the dynamic security policies can be integrated together for ensuring better security systems. Several cloud-based, fog-based, centralized or decentralized access control works are included in this earlier survey in the Sensors journal: A Survey of Context-Aware Access Control Mechanisms for Cloud and Fog Networks: Taxonomy and Open Research Issues. Some important data security and role-based access control papers that are relevant in the system security direction: Role-based access control: A multi-dimensional view (a very interesting and earlier paper, the authors can have a look RS Sandhu’s recent papers); Enforcing role-based access control for secure data storage in the cloud; The privacy-aware access control system using attribute-and role-based access control in private cloud; Classification of personal data security threats in information systems; Mobile Data Security Detection Technology Based on Machine Learning; This paper is an interesting paper that covered some early security models: ICAF: a context-aware framework for access control. The authors can have a look their recent papers as well.
  • An important section is missing before conclusion. It is necessary to have a section titled 'Future Research Issues' and list relevant issues following the research challenges that discussed in the earlier survey sections.
  • The final section could be seen as 'Conclusion and Future Research Directions'. Please provide guidance to direct the listed research issues here so that future scholars can continue their research following your guidance.

Author Response

(The authors gave the same response as above.)

Round 2

Reviewer 2 Report

As a general audience, I still believe the 'data system security' and "data security and access control mechanisms" are very interrelated areas. The following comments might be considered to improve the paper that have been mentioned earlier.

  • It is necessary to consider 'Data Security and Access Control' direction and cover relevant papers. A few of the important papers can be included here. It would be great if the authors could discuss dynamic security policies (e.g., dynamic context-aware access control policies – cloud-based, fog-based, centralized or decentralized, based on the domains) and provide several future research directions, especially focusing – how the data security systems and the dynamic security policies can be integrated together for ensuring better security systems. Several cloud-based, fog-based, centralized or decentralized access control mechanisms are included in this earlier survey in the Sensors journal: A Survey of Context-Aware Access Control Mechanisms for Cloud and Fog Networks: Taxonomy and Open Research Issues. Some important data security and role-based access control papers that are relevant in the system security direction: Role-based access control: A multi-dimensional view (a very interesting and earlier paper, the authors can have a look RS Sandhu’s recent papers); Enforcing role-based access control for secure data storage in the cloud; The privacy-aware access control system using attribute-and role-based access control in private cloud; Classification of personal data security threats in information systems; Mobile Data Security Detection Technology Based on Machine Learning; This paper is an interesting paper that covered some early data security models. 
  • An important section is missing before the Conclusion. It is necessary to have a section titled 'Future Research Issues' and list relevant issues following the research challenges that discussed in the earlier survey sections.
  • The final section could be seen as 'Conclusion and Future Research Directions. Please provide guidance to direct the listed research issues here so that future scholars can continue their research following your guidance.

Author Response

Dear all,

In response to your last review and following your suggestions,  the subsequent changes were made to this piece of literature:

- The issue ‘Data Security and Access Control ‘ was included in an extensive paragraph on the theme, based on fresh and latest discussions  (Kayes et al. 2020);

-   The heading of the Conclusion section was modified;

-  A new section of Future Research Issues was also added.

Many thanks in advance,

Ricardo Jorge Raimundo

Albérico Rosário
